# Indirect Somatic Embryogenesis and Cryopreservation of *Agave tequilana* Weber Cultivar ‘Chato’

**DOI:** 10.3390/plants10020249

**Published:** 2021-01-28

**Authors:** Lourdes Delgado-Aceves, María Teresa González-Arnao, Fernando Santacruz-Ruvalcaba, Raquel Folgado, Liberato Portillo

**Affiliations:** 1Centro Universitario de Ciencias Biológicas y Agropecuarias, Universidad de Guadalajara, Zapopan 45200, Mexico; bmlda108@gmail.com (L.D.-A.); fernando.santacruz@academicos.udg.mx (F.S.-R.); 2Laboratorio de Biotecnología y Criobiología Vegetal, Facultad de Ciencias Químicas, Universidad Veracruzana, Orizaba 94340, Mexico; teregonzalez@uv.com.mx; 3Huntington Library, Art Museum, and Botanical Gardens, San Marino, CA 91108, USA; rfolgado@huntington.org

**Keywords:** regeneration, picloram, cryoplate, vitrification solutions, long-term preservation

## Abstract

*Agave tequilana* Weber cultivar ‘Chato’ represents an important genetic supply of wild severely in decline populations of ‘Chato’ for breeding and transformation programs. In this work, the indirect somatic embryogenesis and cryopreservation of Somatic Embryos (SEs) were investigated using the ‘Chato’ cultivar as a study case. Methods: Embryogenic calli were induced by the cultivation of 1 cm of young leaves from in vitro plants on MS semisolid medium supplemented with 24.84, 33.13, 41.41, 49.69, and 57.98 μM 4-amino-3,5,6-trichloro-2- pyridinecarboxylic acid (picloram) in combination with 2.21, 3.32, and 4.43 μM 6-benzylaminopurine (BAP). The origin and structure of formed SEs were verified by histological analysis. Cryopreservation studies of SEs were performed following the V-cryoplate technique and using for dehydration two vitrification solutions (PVS2 and PVS3). Results: The highest average (52.43 ± 5.74) of produced SEs and the Embryo Forming Capacity (estimated index 52.43) were obtained using 49.69 µM picloram and 3.32 µM BAP in the culture medium. The highest post-cryopreservation regrowth (83%) and plant conversion rate (around 70%) were achieved with PVS2 at 0 °C for 15 min. Conclusion: Our work provides new advances about somatic embryogenesis in *Agave* and reports the first results on cryopreservation of SEs of this species.

## 1. Introduction

*Agave* plants are distributed in several wild and cultivated areas of different Mexican states, as well as preserved by many local human populations [1]. There are various cultivars of *A. tequilana* that were previously used to make tequila, among which are ‘Azul’, ‘Chato’, ‘Chino’, ‘Pata de mula’, ‘Mano larga’, ‘Bermejo’, ‘Xigüin’, and ‘Moraleño’ [2,3]. Among them, *A. tequilana* ‘Chato’ is a valuable resource, of which wild populations are severely displaced due to overexploitation of specific cultivar (*A. tequilana* ‘Azul’ monocrops). The growing demand for products derived from *Agave* spp. increases the need for plantations; however, most species of economic interest are severely affected at the risk of loss by the intense pressure and use of these plant natural resources. Besides, the slow growth to reach its sexual reproductive stage (8–15 years) has made this resource particularly vulnerable since the plants are usually exploited before the formation of the flower stalk, avoiding the dissemination of seeds and reducing variability [4,5]. *Agave* plant propagation is primarily achieved by the multiplication of rhizomatous shoots, which arise from the basal stem of parent plants. *A. tequilana* ‘Chato’ is considered a key cultivar to provide variability to species of commercial use and contributes to the diversify of their populations [6]. However, *A. tequilana* ‘Azul’ is the only approved cultivar by the Mexican regulation to produce tequila [7], and therefore, the only one with the protection of origin denomination [6,8]. This implies that the tequila industry ignores the genetic resources of other important local cultivars. 

The application of tissue culture techniques in *Agave* spp. has been useful to promote large-scale plant production of endangered and economically important species [9,10,11,12,13,14,15,16,17]. Somatic embryogenesis represents a valuable in vitro regeneration system [14] to produce somatic embryos (SEs), which are essential for breeding and genetic transformation programs, as well as for germplasm conservation. In addition, the histological studies represent an important complementary approach to follow up and verify the course of the embryogenic process from the cellular origin of SE until germination of the seedling [18]. The developmental stages of the unicellular origin of somatic embryogenesis for *A. tequilana* have been previously presented [14].

SEs as biotechnological products can be long-term preserved only by using cryopreservation techniques [19]. The cryogenic storage (i.e., preservation at ultralow temperature, mainly in liquid nitrogen (LN), −196 °C) of complex structures with a heterogeneous cellular composition like somatic embryos has been mostly achieved using different vitrification-based procedures [20,21]. A common characteristic of these techniques is that dehydration prior to cooling is the critical step to induce vitrification, which means the transition of the aqueous contents of tissues directly to the amorphous glass phase during the rapid or ultrarapid immersion in LN. The development of the V-cryoplate method [22], which involves the encapsulation of plant material over aluminum cryoplates, allowing the manipulation of many samples at the same time along the different stages of the protocol. In addition, it ensures ultrarapid rates of cooling and warming, which help to improve the post-cryopreservation recovery and, consequently, the effectiveness of the procedure [22,23]. Following the V-cryoplate approach, samples are osmotically dehydrated by exposure to Plant Vitrification Solutions (PVS), which are highly concentrated mixes of cryoprotectants. PVS2 formulation [24] has proved to be the most effective for dehydration of the tissue of different plant species, while PVS3 has been demonstrated to be less toxic and very convenient when PVS2 results cytotoxic [25].

The integration of somatic embryogenesis and cryopreservation has been successfully achieved in some species, such as olive [26], cocoa [27], and avocado [28]. However, so far, there are no reports of its application to *Agave* embryogenic cultures. 

This work aimed to induce the somatic embryogenesis in *A. tequilana* ‘Chato’, validate the embryogenic process by histological analysis, and study the cryopreservation of agave SEs following the V-cryoplate procedure.

## 2. Results

### 2.1. Indirect Somatic Embryogenesis

Produced calli formed clumps and were creamy in color. The embryogenic calli were friable and contained numerous elongated, spherical units that formed translucent and immature SEs. The results obtained under the factors analysis that included the auxin picloram and the cytokinin 6-benzylaminopurine (BAP) showed a highly significant difference as well in their interaction (*p* < 0.01). Since an interaction was detected between factors, all growth regulator combinations were analyzed and were found to have a significant effect (*p* < 0.01) on the differentiation response of calli to produce SEs. Table 1 shows the least significant difference (LSD) test for the mean number of somatic embryos (*p* < 0.05). The significantly highest number of SEs (52.43 ± 5.74 SEs) was achieved by adding 49.69 µM picloram and 3.32 µM BAP to the induction culture medium. The formation of SEs significantly decreased using other combinations. The same concentration (49.69 µM) of picloram gave the best result (24 SEs) when used with the lowest (2.21 µM) concentration of BAP. By contrast, the highest concentration of BAP only had a minor effect (16 SEs) at a lower picloram concentration (33.13 µM). The Embryo Forming Capacity (EFC) of callus from 1 cm^2^ of young leaves derived from in vitro plants of ‘Chato’ cv. ranged from 1.29 up to a maximum value of 52 after 60 days of culturing at the best combination of picloram (49.69 µM) with BAP (3.32 µM).

The first step of the embryogenic process was the cellular disorganization of the leaf (Figure 1a,b), followed by the formation of abundant calli (Figure 1c), and the asynchronous proliferation of SEs on the embryogenic callus surface (Figure 1d). The typical structures of the developmental phases of SE were observed: globular (Figure 1e,f), scutellar (Figure 1g,h), and coleoptilar (Figure 1i,j). The maturation of SEs from the globular stage to germination and root development (Figure 1e–p) showed a transition time of approximately eight weeks. The conversion rate of SEs to ex vitro (acclimated) plants (Figure 1q) was 92%.

### 2.2. Histological Observations 

The staining used in the present study characterized embryogenic calli by discriminating small dividing cells with a higher proportion of nucleic acids (stained with acetocarmine due to their affinity to this pigment) from those large and vacuolated (stained with Evan’s blue due to the basic pH). The unicellular origin of the SEs is indicated by an initial asymmetrical cell division (Figure 2a). Pre-embryogenic cells showed properties that are common to cells in the division stage with high metabolic activity (Figure 2b). A suspensor type structure independent of the embryogenic callus could be observed in the next stage (Figure 2c,d). At this stage, we also observed a potential hypophysis like structure, which is a prominent cell zone (cells in contact with the embryo that link it to the suspensor) that promotes the formation of the radicle (Figure 2c). At the globular stage, the delimiting protoderm between the callus and the somatic embryo marked the independence of the embryo (Figure 2e). Then, differentiated pro-vascular strands were observed in scutellar embryos (Figure 2f). At the scutellar stage (Figure 2g,h), the apical axis or plumule (shoot) and basal axis that gives rise to radicle (root) differentiation (Figure 2j). The coleoptilar stage (Figure 2i,j) is characterized by the histodifferentiation of coleoptiles as the last major morphogenic transition of the embryos.

### 2.3. Cryopreservation by the V-cryoplate Method

Somatic embryogenesis is an asynchronous process that makes difficult the selection of a specific embryo size and of a determined physiological stage at a given culture time. After 60 days time on expression medium, we found that the most frequent sizes of SEs, ranged from 0.1 to 4.0 mm, and that globular and coleoptilar were the developmental stages mostly observed. Therefore, based on the higher frequency of these two factors, we selected SEs of 1.0 to 2.0 mm size at the coleoptilar stage to standardize the material choice for cryopreservation. SEs at earlier stages were subcultured on solid MS medium to stimulate their growth until obtaining material according to the pre-established parameters for cryogenic experiments.

Results of cryopreservation experiments expressed by the regrowth and plant conversion of SEs before and after immersion in liquid nitrogen (LN) are shown in Figure 3 and Figure 4. Before cryopreservation, the regrowth of SEs was significantly (*p* ≤ 0.05) influenced by the duration of exposure to both PVSs (PVS2 and PVS3) (Figure 3). Regrowth was significantly (*p* ≤ 0.05) reduced (from 96% to 73%) after 30 min of exposure to PVS2 or 45 min exposure to PVS3 (reduction from 100% to 57%). However, conversion to plants was significantly affected after 15 min of treatment with either of the two PVSs used (Figure 4). After cryopreservation (+LN), regrowth and plant conversion were only detected in SEs that had been dehydrated with either PVS2 or PVS3. In general, SEs of *A. tequilana* ‘Chato’ tolerated all the dehydration durations assessed at low temperature (0 °C) with both PVS. The highest and significantly different percentages of regrowth and conversion to the plant were achieved after 15 min (83% and 73%, respectively) or 30 min (77% and 67%, respectively) of exposure to PVS2 and after 30 min (80% and 70%, respectively) to PVS3. Therefore, to achieve the best post-cryopreservation recovery expressed by the regrowth and plant conversion rates, a longer exposure time to PVS3 was required in comparison to that needed when PVS2 was used. Nevertheless, SEs treated with PVS3 showed faster regrowth during the first 30 days of reculture than those SEs treated with less or for the same time with PVS2. After 60 days of reculture, a similar appearance in plants cryopreserved was reached by all recovered SEs regardless of the PVS applied for dehydration.

Plants formed of cryopreserved SEs displayed normal growth and development (Figure 5), i.e., they were morphologically similar to control plants not subjected to cryopreservation and originating by germination of zygotic (Figure 5a) or somatic (Figure 5b,c) embryos, respectively.

## 3. Discussion

### 3.1. Indirect Somatic Embryogenesis

There are several factors involved in the acquisition of the embryogenic competence in explants cultivated in vitro, such as the balance of hormones, osmotic conditions, change of pH, concentrations of amino acids and salts, and treatments with various chemical substances [29,30,31]. The efficiency of induction depends not only on the culture conditions but also on the genotype, explant source, and its stage of development [32]. In *Agave* spp., somatic embryogenesis has been induced using different types of explants: roots [33], leaf [14], and putative basal part of the stem [34,35]. Regarding growth regulators, 2,4-D (2,4-diclorophenoxiacetic acid) was the most employed auxin in combination with other hormones to induce this process in several species: *A. victoria-reginae* [36], *A. sisalana* [13], *A. tequilana* ‘Azul’ [14], *A. vera-cruz* [37], *A. salmiana* [38], and *A. angustifolia* [17,39].

In our experiment, we replaced the use of 2,4-D with picloram, which in combination with BAP, allowed the indirect development of SEs in *A. tequilana* Weber cultivar ‘Chato’ at a similar rate as previously reported with 2,4-D [14]. Following this approach and under the best-determined culture conditions, the embryogenic calli reached an EFC of 52 with a high frequency of plant conversion (90%). On the other hand, Santiz et al. [40] reported a higher EFC index in *A. grivalgensis* when the concentration of cytokinin was higher than that of the auxin by combining BAP and 2,4-D. In contrast, the balance of auxin-cytokinin concentrations assayed in our study improved the callus embryogenic capacity when BAP was used at a lower concentration than picloram auxin (Table 1). The beneficial effect of picloram to induce somatic embryogenesis has also been demonstrated in the *A. americana* species [35], as well as in other plant species like *Urochloa* [41], *Gasteria verrucosa*, and *Haworthia fasciata* [42]. However, to our knowledge, picloram has not been used before to induce somatic embryogenesis from agave leaf explants.

The morphological evaluation performed during the different stages of development of the whole somatic embryogenesis process (Figure 1) demonstrated accordance with the one reported by Portillo et al. [14], who also used young leaf explants to generate SEs. In addition, it had similarities with the stages observed during the morphological development of zygotic embryos (ZE) of this species (Figure 1p,q). Furthermore, it was visually defined that germinated SEs produced normal plantlets, which resembled the ones obtained by germination of ZE. These observations are contrary to that obtained by Monja-Mio and Robert [34] in *A. fourcroydes* Lem., who reported having achieved direct somatic embryogenesis using the same hormone combination with other explant type; however, the morphology of their developing somatic embryos did not resemble the zygotic ones.

Regeneration systems for *Agave* spp. have not yet been characterized in detail. Somatic embryogenesis and organogenesis have been confused as the same processes [13,34,35,36,37]. Therefore, we emphasize the importance of ontogenetic observations by performing a comparison between SE and ZE of the studied species.

### 3.2. Histological Analysis

In this study, we have documented the unicellular origin and cellular structures (protoderm, scutellum, and coleoptile) in agave embryogenic development, which suggests crucial evidence for the regeneration via somatic embryogenesis versus organogenesis. Embryogenic cells from which embryoids are visibly derived shown a series of common characteristics as a high nucleus-cytoplasm ratio, thick cell wall, and small size (Figure 2a) resembling rapidly dividing meristematic cells (Figure 2b,c). According to Williams and Maheswaran [43], there are several points for discussion among the known regeneration systems, including the unicellular or multicellular origin, the physical or physiological independence of the starting embryogenic cells from the tissue of origin, the similarity or dissimilarity with zygotic embryogenesis, and the induction process controlled by exogenous growth regulators as opposed to the internal physiological state of the tissue explant.

Auxin concentrations higher than cytokinins (i.e., 9.0 or 13.6 μM 2,4-D, and 4.0 μM BAP) have resulted in somatic embryogenesis in *A. tequilana* [14]. It was also observed that a concentration of auxin higher than cytokinin correlates with the unicellular origin (Figure 2a), and the independence of SEs from the parent callus (Figure 2d,e), and its similar development to ZE (Figure 2j), as reported by Portillo et al. [14] and Ayala-González et al. [44]. It has been determined that regeneration can be achieved from meristemoids (organogenesis) when new shoots are induced from callus or directly upon explant tissues, or via SEs that resemble seed embryos developing to seedlings in the same way [43,45]. At this point, the question of the origin of one or several cells for SE is directly related to the coordinated behavior of neighboring cells as a morphogenetic group [43,45]. According to several authors, a somatic embryo is defined as a new individual that arises from a single cell and has no vascular connection to the parent cells; multicellular origin seems to produce embryoids fused with parent cells over a wide area of the root pole of the axis region [13,46], while a unicellular origin is more likely to produce individual embryoids with a narrower structure similar to a suspensor [30,43,45,46,47]. Therefore, the regeneration processes must be thoroughly studied in order to clarify and define whether a multicellular regenerate is a SE or an organ primordium. Halperin, emphasized-the use of histological sections in regeneration systems to determine the cellular origin, morphology, and stages of embryonic development [48]. By means of a histology study and morphology comparison, we found that in our system *Agave* SEs had a unicellular origin in agreement with previous work [14] (Figure 2a), and their development acquired in later stages were congruent (Figure 5a,b) [49].

### 3.3. Cryopreservation by V-cryoplate Method

Dehydration with PVS proved to be a critical step for cryopreservation of the agave SEs. The comparison of the two PVS (PVS2 and PVS3) showed that at the best exposure time, regrowth (about 80%) and plant conversion rate (about 70%) remained similar before and after cryopreservation, indicating that the immersion in LN caused no additional detrimental effects.

The positive effect of using a low temperature (0 °C) for dehydration with a PVS was first reported by Yamada et al. [50]. This approach of osmoprotection has been very useful with other tropical species, which usually are relatively sensitive to such dehydration treatments [51]. At this low temperature, PVS2 was demonstrated to be more effective than PVS3 in a shorter dehydration period. It was convenient to reduce the harmful effects due to the overexposure of tissues to PVS2. This effectiveness seems to be related to its chemical composition and lower viscosity in comparison with PVS3, which allows removal of the freezable water from cells [52] and increase the ability to vitrify during rapid immersion in LN [53].

Some studies on cryopreservation of SEs of avocado [54] and olive [55] have reported 30 min of dehydration as the optimal exposure time to PVS2 using a droplet-vitrification procedure. There are other cases in which more prolonged exposures (60–90 min) have proved to be beneficial [26,28,53]. However, our results showed that in the case of the SEs of cultivar ‘Chato’, increasing the exposure time beyond 30 min was detrimental, whichever PVS was used, provoking a significant drop in the regeneration response. 

In this work, we determined suitable conditions to cryopreserve SEs of *Agave tequilana* cultivar ‘Chato’ following the V-cryoplate procedure and using two PVS (PVS2 or PVS3) up to 30 min at 0 °C. So far, the use of the V-cryoplate technique in SE has only been reported by Pettinelli et al. [56] to cryopreserve SEs derived from in vitro roots of guinea (*Petiveria alliacea*) and dehydrated with PVS2 for 15 min. These results match with the duration of osmoprotective treatment, which allowed them to obtain the best response after cryopreservation of agave SEs. Future adaptation of this protocol to other agave cultivars will depend on the water content and the sensitivity of their SEs to the PVS.

## 4. Materials and Methods

### 4.1. Induction of Indirect Somatic Embryogenesis

Rhizomatous shoots of *Agave tequilana* Weber cultivar ‘Chato’ were obtained from parent plants at physiological maturity (seven years old), which were provided by The Botanical Garden from the University of Guadalajara for initial in vitro shoot cultures [14]. Micropropagation of rhizomatous shoots was carried out in MS medium [57] supplemented with L2 vitamins [58] and 22.15 μM of 6-benzylaminopurine (BAP). The genotype 7 (SEC7) was selected due to its high frequency of proliferation of vigor shoots. The somatic embryogenesis process was followed and evaluated, defining three stages: callus induction, callus differentiation to SE, and conversion of obtained SEs to plants. For callus induction, segments (1 cm^2^) adjusted with a millimetric ruler of young leaves from in vitro plants were cultivated for 40 days in a glass jar of 100 mL capacity with 25 mL of culture medium. The induction culture media comprised the MS basal formulation supplemented with 24.84, 33.13, 41.41, 49.69, or 57.98 μM of 4-amino-3,5,6-trichloro-2-pyridinecarboxylic acid (picloram) in combination with 2.21, 3.32, or 4.43 μM BAP (a 5 × 3 bifactorial design), 3% sucrose (27360 Golden Bell^MR^), and solidified with 8 g L^−1^ agar (A-1296 Sigma^®^). Four leaf segments were placed per culture jar (experimental unit) and for each combination of growth regulators. The experiment was replicated four times. 

To induce callus differentiation (second stage), the masses of calli produced were transferred to Petri dishes with 25 mL of expression medium and cultured for 60 days. Expression medium consisted of a modified MS basal formulation [59] supplemented with 500 mg L^−1^ glutamine, 250 mg L^−1^ casein hydrolysate, 3% sucrose, and solidified with 6 g L^−1^ phytagel (P-8169 Sigma^®^) [14].

The pH of all culture media was adjusted to 5.8 ± 0.05, and then, they were sterilized in an autoclave at 121 °C with 1.3 kg cm^−2^ of pressure for 15 min. Cultures conditions for callus induction and differentiation into somatic embryos comprised the exposure of samples to a photoperiod (16/8 h light/dark), with a luminous intensity of 27 μmol m^−2^ s^−1^, and the regulation of temperature at 25 ± 2 °C. 

The number of generated SEs per callus was recorded after 60 days of culturing using a stereoscope with 10× magnification (Leica^®^ microsystems EZ4 W). For all treatments, the means of percentages of embryogenic calli (which formed SEs) and from the number of formed SEs were estimated considering four explants per Petri dish. An index of *Embryo Forming Capacity* (EFC) was defined by adapting and modifying the equation previously reported [60].
EFC = (Average percentage of calli forming SE) (Average number of formed SE)/100

The final stage of the somatic embryogenesis process was performed by transferring one hundred SEs from the expression medium to MS medium without growth regulators for the other 60 days. For acclimation, rooted plantlets were then removed from the culture medium and placed in trays with a wet soil mixture of 7:3 (*v*/*v*) peat moss and perlite under greenhouse conditions with full sun at 27 ± 5 °C and 75% RH. The conversion rate of SEs to plants was calculated after the 60 days of culture on MS medium when the material was ready to be ex vitro transferred to be acclimated.

### 4.2. Histological Analysis

Histological studies were carried out to support the theories on the unicellular and multicellular origin, development, and characteristics of SE according to [61,62]. Fresh embryogenic calli (0.02 g) and SEs of the genotype SEC7 at different developmental stages were fixed using 70% *v*/*v* alcohol and embedded in polyethylene glycol (PEG, 1450 molecular mass) in a 1:4 proportion (PEG: deionized water) according to the protocol described [63]. The experiments were replicated six times. A rotatory microtome was used to obtain 15 μm sections from the samples in PEG; then, they were stained with a double treatment using acetocarmine 0.5% (1:1 *w*/*v*) and 0.5% Evan’s blue (1:1 *w*/*v*) [64]. A light microscope was used to analyze the tissues. 

### 4.3. Cryopreservation of Somatic Embryos

Experiments were performed following the V-cryoplate method [65] and using cryoplates with ten wells with oval shapes according to design No. 3 (37 mm length × 0.5 mm thickness, wells with 2.5 mm length, 1.5 mm width, and 0.75 mm depth), manufactured by the Japanese Company (Taiyo Nippon Sanso Corp; Tokyo, Japan). SEs (1–2 mm length) were precultured for 1 day on MS solid medium with 0.3 M sucrose in the dark, followed by their transfer, one by one, to the wells of the cryoplates where they were encapsulated. First, embryos were covered with 2.5 μL of sodium alginate solution (2% *w*/*v*) containing 0.4 M sucrose, and then, with calcium chloride solution (0.1 M) gently added to allow polymerization of calcium alginate. After 15 min, calcium chloride solution was removed, and cryoplates with the encapsulated SEs were exposed to loading solution containing 1 M sucrose and 2 M glycerol for 15 min at room temperature, followed by the dehydration with a PVS solution pre-cooled in an ice bath. The effect of two vitrification solutions was evaluated: PVS2 (30% *v*/*v* glycerol, 15% *v*/*v* dimethyl sulfoxide, 15% *v*/*v* ethylene glycol, and 13.7% *w*/*v* sucrose) [24] and PVS3 (50% *v*/*v* glycerol and 50% *w*/*v* sucrose) for 0, 15, 30, 45, and 65 min prior (−LN) and after (+LN) direct immersion in LN [66]. Warming took place at room temperature using liquid MS as recovery medium with 1.2 M sucrose, where the cryoplates with samples were immersed for 15 min. Subsequently, the calcium alginate gel remaining in the SEs was carefully removed before the recovery culture. Post-cryopreservation recovery took place by transferring the embryos to semisolid MS medium supplemented with 0.3 M sucrose and culturing them for seven days in darkness at 25 °C, followed by the reculture onto semisolid MS medium and exposure to photoperiod. After cryopreservation, regrowth of SEs was evaluated after 45 days of culturing on MS medium, expressed by the elongation of coleoptile and the formation of radicles. The conversion rate of cryopreserved SEs to plants was determined after additional 60 days of culture in the MS medium (a total culture period of 105 d), and the morphological development of the obtained plants was compared with that of plantlets derived of non-cryopreserved SE and of ZE germinated in vitro. Non-cryopreserved controls were cultivated for 60 days in MS medium for the comparison.

### 4.4. Statistical Analysis

The experimental design was completely randomized to study both the embryogenesis and cryopreservation processes. Somatic embryogenesis experiments were replicated four times using four explants per Petri dish, and the results were expressed as the average ± standard deviations.

Cryopreservation experiments were replicated three times using ten SEs per replicate. Dependent variables were regrowth and the conversion rate of SEs to plant before (−LN) and after (+LN) the immersion in LN. 

Results of somatic embryogenesis assays were analyzed by two-way analysis of variance (ANOVA) (levels of picloram and BAP). Results of cryopreservation assays were processed by one-way ANOVA (PVS). Means were compared by the least significant difference (LSD) range test with an error rate at *p* ≤ 0.05. All statistical analyses were carried out using the Minitab^®^ statistical software 17.2.1.

## 5. Conclusions

In this work, we reported the efficient application of in vitro techniques to induce indirect regeneration of SEs from *Agave tequilana* cultivar ‘Chato’ and their subsequent cryopreservation. The addition of growth regulators picloram and BAP to MS semisolid medium proved to be useful to induce somatic embryogenesis and obtain large amounts of actively growing embryos from callus derived of in vitro cultured leaf explants. The histological analysis illustrated this process and supported the single-cellular origin of SEs. The V-cryoplate method resulted in a practical and effective approach to cryopreserve SEs of cultivar ‘Chato’ using two PVS (PVS2 or PVS3). The experimental findings reported here represent viable alternatives to generate and safely store material for the long-term, which can be a new source of material for the commercial propagation of this plant species, the production of elite lines, and of usefulness for genetic transformation programs. This work provides new advances about somatic embryogenesis in *Agave* spp. and reports the first results on cryopreservation of SE of this species.

## Figures and Tables

**Figure 1 plants-10-00249-f001:**
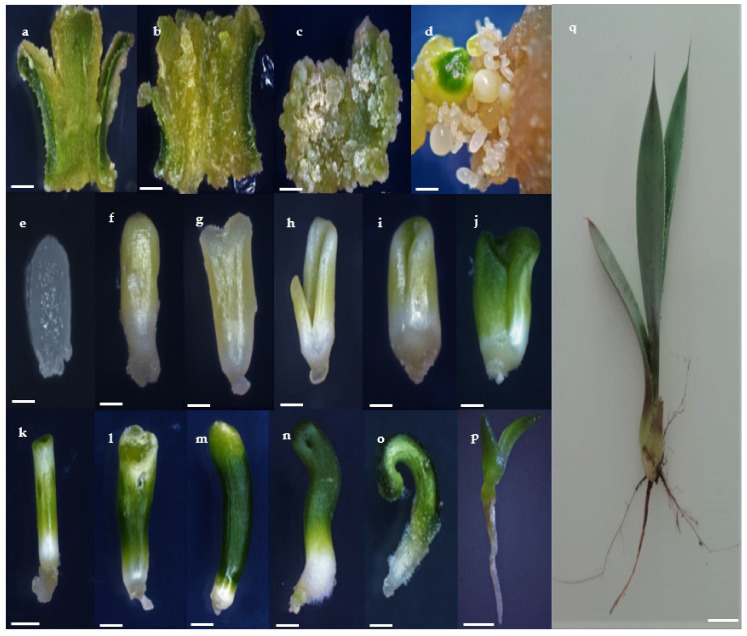
Stages of indirect somatic embryogenesis in *Agave tequilana* Weber cultivar ‘Chato’. Callus production in leaf after (**a**) 10 days, (**b**) 25 days, and (**c**) 40 days of culturing (bars 2.0 mm), (**d**), embryogenic callus in expression medium with somatic embryos (SEs) at several stages after 60 days of culturing (bar 2.0 mm), (**e**) and, (**f**), globular SEs without and with chlorophyll presence (bars 0.2 mm and 0.5 mm), respectively, (**g**) and (**h**) scutellar stage (bars 0.5 and 1.0 mm), respectively, (**i**–**o**), subsequent development of the colleoptilar stage presenting radicle origin. (**i**,**j**) (bars 1.0 mm and (**k**–**o**) bars 2.0 mm), respectively, (**p**) Somatic embryo elongation and tissue maturation (bar 5.0 mm), and (**q**) ex vitro-grown plant under greenhouse conditions (250 d) (bar 1.0 cm).

**Figure 2 plants-10-00249-f002:**
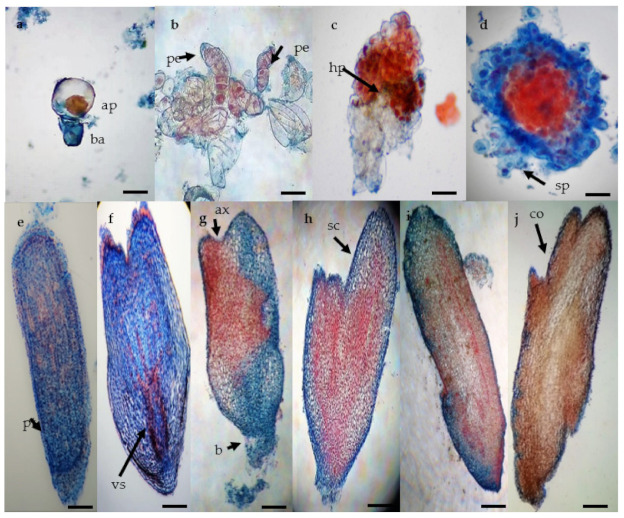
Histological analysis of somatic embryogenesis process in *Agave tequilana* Weber ‘Chato’. (**a**) asymmetric cell division giving rise to small apical cell (red) and basal cell (blue) (bar 50 µm), (**b**) proembryos showing group of cells with one side of rapid cellular division (black arrow) (bar 100 µm), (**c**) late proembryo stage with dyed suspensor in blue and hypophysis (bar 100 µm), (**d**) specific formation pattern of the globular stage with vestigial suspensor structures (bar 100 µm), (**e**) late globular stage with the formation of the protoderm (bar 200 µm), (**f**) late globular stage (bar 200 µm), (**g**) early scutellar stage showing an apical axis (black arrow) (bar 200 µm), (**h**) late scutellar stage (bar 200 µm), (**i**) early coleoptillar stage (bar 200 µm), (**j**) late colleoptillar stage (bar 200 µm). Apical cell (ap), basal cell (ba), proembryo (pe), hypophysis (hp), suspensor (sp), protoderm (pt), pro-vascular strands (vs), apical axis (ax), basal axis (bx), scutellum (sc), and coleoptile (co).

**Figure 3 plants-10-00249-f003:**
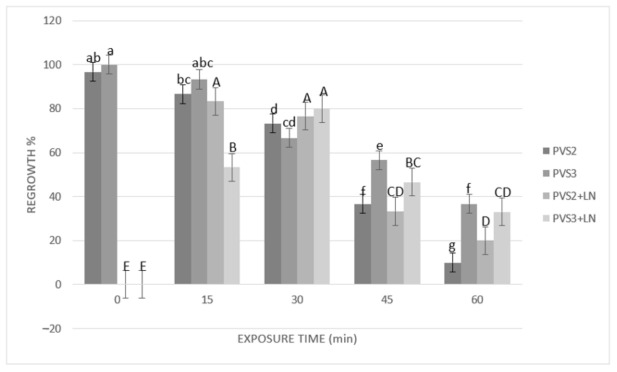
Effect of exposure length to plant vitrification solutions (PVS) PVS2 or PVS3 at 0 °C on regrowth of somatic embryos (SEs) before and after cryopreservation. SEs were precultured on MS solid medium with 0.3 M sucrose in the dark for one day, encapsulated over the cryoplate with calcium alginate (2%) containing 0.4 M sucrose, and loaded with 1 M sucrose + 2 M glycerol for 15 min before exposure to PVS. Data are represented by means (%) ± standard error (se). Bars with the same letter are not significantly different *p* ≤ 0.05 (LSD test). Lowercase letters refer to non-cryopreserved controls, and uppercase letters refer to cryopreserved plus liquid nitrogen (+LN) samples. Regrowth was detected as the percentage of SEs that showed elongation of coleoptile and the formation of radicle 45 days after their transfer to the culture medium.

**Figure 4 plants-10-00249-f004:**
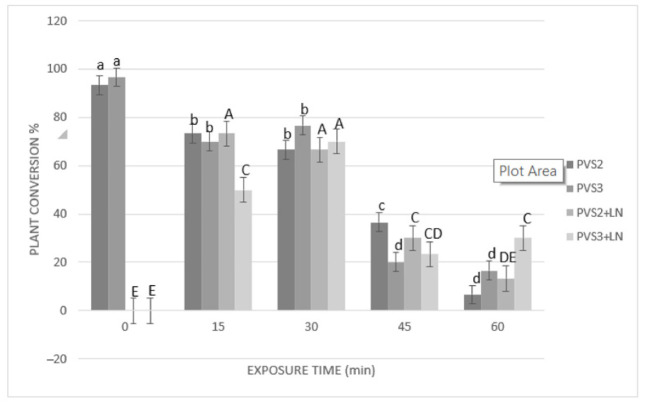
Effect of exposure length to PVS2 or PVS3 at 0 °C on the conversion of somatic embryos (SEs) to plants before and after cryopreservation. SEs were precultured on MS solid medium with 0.3 M sucrose in the dark for one day, encapsulated over the cryoplate with calcium alginate (2%) containing 0.4 M sucrose, and loaded with 1 M sucrose + 2 M glycerol for 15 min before exposure to PVS solution. Data are represented by means (%) ± standard error (se). Bars with the same letter are not significantly different *p* ≤ 0.05 (LSD test). Lowercase letters refer to non-cryopreserved controls, and uppercase letters refer to cryopreserved (+LN) samples. Conversion to plants of cryopreserved SEs was evaluated after 105 days of culturing on MS medium.

**Figure 5 plants-10-00249-f005:**
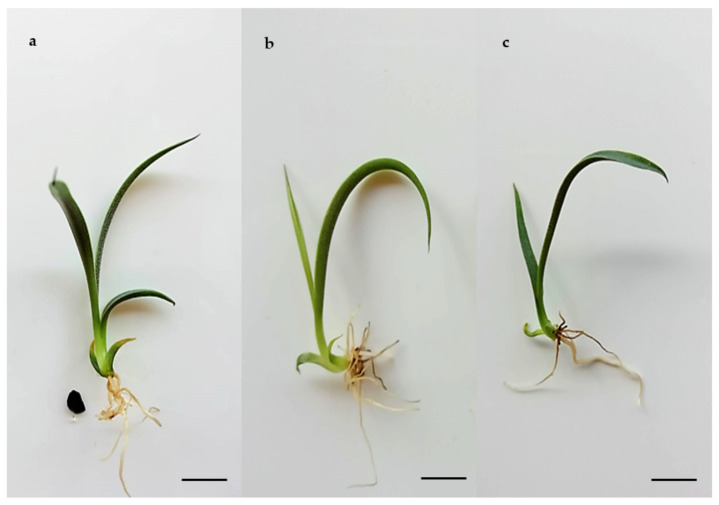
Conversion to plants of *Agave tequilana* Weber ‘Chato’. (**a**) Plant derived from zygotic embryo after 60 days of culture on MS medium (bar 2 cm), (**b**) Plant derived from non-cryopreserved SE after 60 days of culture on MS medium (bar 1 cm), (**c**) Plant derived from SE recovered after cryopreservation and cultivated for 105 days on MS medium (bar 1 cm).

**Table 1 plants-10-00249-t001:** Effect of plant growth regulators on the differentiation rate of somatic embryos derived from calli of in vitro young leaves of *Agave tequilana* ‘Chato’.

Treatment	Concentration of Growth Regulators (µM)		Calli that Formed SEs %	Number of Somatic Embryos (Mean ± se) *		EFC
	Picloram	BAP				
1	24.84	2.21	56.25	10.93 ± 4.58	fg	6.15
2	33.13	2.21	68.75	16.18 ± 6.30	def	11.12
3	41.41	2.21	31.25	10.81 ± 9.17	fg	3.37
4	49.69	2.21	56.25	24.12 ± 4.00	bc	13.57
5	57.98	2.21	43.75	20.25 ± 5.61	cde	8.85
6	24.84	3.32	31.25	5.06 ± 2.29	g	1.58
7	33.13	3.32	62.50	13.37 ± 5.34	ef	8.35
8	41.41	3.32	93.75	29.43 ± 5.30	b	27.59
9	49.69	3.32	100.00	52.43 ± 5.74	a	52.43
10	57.98	3.32	75.00	21.87 ± 6.19	bcd	16.40
11	24.84	4.43	56.25	27.18 ± 2.33	bc	15.29
12	33.13	4.43	43.75	16.00 ± 4.15	def	7.00
13	41.41	4.43	31.25	4.56 ± 2.27	g	1.42
14	49.69	4.43	25.00	5.18 ± 1.40	g	1.29
15	57.98	4.43	37.50	9.37 ± 8.92	fg	3.51

* Data represent the means ± standard error (se) of number of somatic embryos (SEs) per explant of 1 cm^2^. Data within columns with the same letter are not significantly different, i.e., *p* < 0.05 (least significant difference (LSD) test). EFC: Embryo Forming Capacity.

## Data Availability

Not applicable.

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
