# Peer review of "Indirect Somatic Embryogenesis and Cryopreservation of Agave tequilana Weber Cultivar ‘Chato’"

_plants, 2021, doi:10.3390/plants10020249_

Round 1
Reviewer 1 Report
The manuscript describes a protocol for the somatic embryogenesis and plant regeneration of an Agave cultivar. Although this protocol might be important to maintain and widen the Agave germplasm used for practical purposes, the scientific novelty of the work is limited. Although the authors emphasize the unicellular origin of the somatic embryos (SE), there is no direct proof for it. Even if embryogenic callus is formed from an individual cell, the unicellular origin of the embryo is unproven. Without the use of any specific marker of embryogenesis and using only acetocarmine staining the origin of the embryos remain unclear. In general, the discussion has several overstatements. The authors should not go too far to draw conclusions. The manuscript should remain at the methodological level. The amalgamation of Results and Discussions would be beneficial.
The text is full of poor sentences.
I attach a corrected version, but a more thorough revision of the style is required.

Reviewer 2 Report
Dear Authors,
You have done extensive work to compile the efficient regeneration procedure via indirect somatic embryogenesis of Agava tequila and propose the effective factors for cryopreservation of this species. The obtain results could interest many researchers and readers. Although the work is interesting, I think that You should take a count small modification of this article. I recommend publishing it in "Plants" after correcting listed below suggestions:
Results
- 2e - in my opinion presented here embryo is of late globular stage
- 5 – should be included in Results chapter
- Line 159-161 – there are information about the morphological similarity of the mather plants and those obtained by conversion from somatic embryos after cryopreservation. there are only pictures, there are no any concrete parameters (eg. heigh, root system, stomata description et cetera) of their morphology. That would be interesting ot compare the histology of SE after cryopreservation and ploidy of mature and regenerated plants and another physiological and/or molecular features.Of course that would be a material for the next article and I encourage the Authors to do it.
Materials and Methods
- Line 292 – University
- Line 317 – „by” cancel
- Line 334 – „by” cancel
- Line 336-337 – please explain the results of this staining
I hope my comments will be helpful.
With best regards,
Reviewer 3 Report
See file.

Author Response
Number of lines did not match after modifications, please see lines in parenthesis
Keywords: picloram word was added
Introduction: A previous study on somatic embryogenesis was included. Induction rate with picloram was included in discussion
Results: The organization of the data reported and tables have new organization.
The interpretation of results on somatic embryogenesis has been now supported with previous work made from one of the authors of this paper [14] stating and describing in detail the unicellular origin of the process.
Figure 2a has been replaced. Caulinar meristem cannot be seen until embryoid germination (as in figure 1p).
Statistical analysis now are more clear explained and interaction of factor is now mentioned.
Line 204 (304): statement on picloram was already modified.
Discussion :
Line 219 (321): As mentioned above, we have found a unicellular origin for Agave tequilana somatic embryogenesis [14], also we have information about the fate cell at molecular level (see Portillo et al., 2012 Protoplasma 249: 1101-1107). Moreover, the reference (Fehér, 2019) mentioned by the referee x also states that “somatic embryos can form from single somatic cell”.
Line 289 (337): We consider a meristemoid as a de novo meristem, as embryoid is used for somatic embryo.
Line 308 (349): Figures 2 and 5 were referred as evidence.
Material and methods:
Line 292 (396): a reference was included
Line 301 (406) : number of explants and treatment were clarified in the text. In previous experiments we have determined that this number of explants offers a solid statistical analysis.
Line 323 (428). Transference medium was defined. It was not necessary to maintain embryogenic capacity since we had enough somatic embryos for experimentation. Did not take place a secondary embryogenesis. Germination of somatic embryos was mentioned in results on cryopreservation.
Line 350 and 352 (456 and 458): References were included
Line 351 (458): Incubation at 0°C was not necessary
Line 359 (459): Recovery medium was defined
The English in the manuscript has been thoroughly checked and edited for language and form.
Round 2
Reviewer 1 Report
I accept the answers to my comments, however, the style is still weak. I attach a version with further recommendations for improvement.

Reviewer 2 Report
I recommend the article titled: Indirect somatic embryogenesis and cryopreservation of Agave tequilana Weber cultivar ‘Chato’" to be accepted in this form in Plants journal.
Reviewer 3 Report
SEE FILE
